# Ex Uno Plures: Splitting One Model into an Ensemble of Subnetworks

## Abstract

Monte Carlo (MC) dropout Gal & Ghahramani (2016) is a simple and efficient ensembling method that can improve the accuracy and confidence calibration of high-capacity deep neural network models. However, MC dropout is not as effective as more compute-intensive methods such as deep ensembles Lakshminarayanan et al. (2016). This performance gap can be attributed to the relatively poor quality of individual models in the MC dropout ensemble and their lack of diversity. These issues can in turn be traced back to the coupled training and substantial parameter sharing of the dropout models. Motivated by this perspective, we propose a strategy to compute an ensemble of subnetworks, each corresponding to a non-overlapping dropout mask computed via a pruning strategy and trained independently. We show that the proposed subnetwork ensembling method can perform as well as standard deep ensembles in both accuracy and uncertainty estimates, yet with a computational efficiency similar to MC dropout. Lastly, using several computer vision datasets like CIFAR10/100, CUB200, and Tiny-Imagenet, we experimentally demonstrate that subnetwork ensembling also consistently outperforms recently proposed approaches that efficiently ensemble neural networks.

## 1 Introduction

An effective way to improve model accuracy and confidence calibration in deep learning is ensembling. One efficient technique that leverages this idea is "Monte Carlo (MC) dropout" Gal & Ghahramani (2016) which extends the popular dropout technique used for regularization during training Srivastava et al. (2014). In MC Dropout, test-time inference involves multiple forward passes through the model, each executed with a different random dropout mask as in during the training phase. This yields an ensemble of predictions which are then averaged.

While MC dropout can improve a baseline model, it is still inferior to explicit ensembles of neural networks trained independently with random initialization (called deep ensembles) Lakshminarayanan et al. (2016). Using the perspective of the error-ambiguity decomposition Zhou (2009), we can attribute this performance gap to the relatively poor performance of individual models and/or limited diversity in the MC dropout ensemble. We further hypothesize these issues are largely due to the extensive parameter sharing among MC dropout models.

With this perspective in mind, we explore the idea of creating an ensemble of subnetworks in which a pre-determined number of non-overlapping dropout masks are used. We present an easy-to-implement greedy optimization procedure that sequentially computes dropout masks via a recent dropout-mask optimization technique and trains each subnetwork independently. The resulting algorithm enables us to obtain a diverse ensemble of non-overlapping subnetworks within one neural network. That is, we are able create many models out of one[1]. We demonstrate that subnetwork ensembling consistently outperforms MC dropout and several other recently proposed approaches that efficiently ensemble neural networks in terms of both accuracy and uncertainty estimates. We also show that our proposed approach achieves results on par with that of deep ensembles, yet with the much better test-time computational efficiency.

---

[1]Hence our title: Ex uno plures.

**Summary of Contributions.**

1. We present the novel idea of ensembling non-overlapping subnetworks within one neural network architecture.

2. We propose a simple sequential pruning based procedure to enhance the performance of subnetwork ensembling.

3. We demonstrate and discuss the regularization effect achieved by training pruned networks and using a randomized and frozen fully connected layer in the network.

4. Our experiments demonstrate that subnetwork ensembling outperforms MC dropout and several state-of-the-art methods for efficient ensembling.

## 2 Related Works

**Ensemble Learning**  An ensemble of models has long known to be an effective method to boost the performance of machine learning models Dietterich (2000); Zhou (2009). More recently, with the growing interest in deep learning, ensembles of neural networks have gained much attention. Notably, Lakshminarayanan et al. Lakshminarayanan et al. (2016) demonstrated that simple ensembles of neural networks (NNs) trained independently (called "Deep Ensembling") can offer improved predictive uncertainty and accuracy. In fact, somewhat surprisingly, deep ensembles often outperform more sophisticated Bayesian NNs. Building on this, several recent works have attempted to understand the unexpected effectiveness of deep ensembles Fort et al. (2019); Lobacheva et al. (2020); Rahaman & Thiery (2020); Wen et al. (2020).

To further enhance the performance of ensembles of neural networks, previous methods have also pinpointed the importance of model diversity Jiang et al. (2017); Krogh & Vedelsby (1994); Zhang et al. (2020), and explored ways to promote diversity in ensembles. For instance, Sinha et al. Sinha et al. (2020) proposed an Information Bottleneck-based approach to explicitly stimulate diversity among predictions. In related work, Jain et al. Jain et al. (2020) utilized out-of-distribution samples to encourage diversity among models. Enhanced diversity can also be obtained by ensembling across different architectures of NNs through neural architecture search Zaidi et al. (2020) or by varying the hyperparameters of models Wenzel et al. (2020).

Our proposed method can be regarded as an approach for efficient ensembles of NNs. Several techniques have been previously proposed in this direction. For instance, BatchEnsemble Wen et al. (2019) makes use of rank-one matrices to approximate weight matrices in NNs for fast ensembling of models. Lately, a multi-input multi-output (MIMO) Havasi et al. (2020) configuration was discovered to be an effective method to utilize a single model's capacity to train multiple subnetworks. Another recent work Durasov et al. (2020), similar to ours, uses a set of pre-determined masks in place of the stochastic sampling of MC dropout for improved uncertainty estimation. However, their masks are randomly generated at the beginning of training and frozen. There is also significant sharing of parameters among the models, potentially reducing diversity and thus leading to sub-optimal performance.

**Neural Network Pruning**  Network pruning aims to compress neural networks by reducing the number of parameters present in the model Frankle & Carbin (2018); Guo et al. (2016); Han et al. (2015a;b); Lee et al. (2018); Li et al. (2016); Liu et al. (2018). It often involves selecting and discarding parameters from a pre-trained network, after which the compressed network is fine-tuned. While earlier approaches choose parameters based on their magnitudes Frankle & Carbin (2018); Han et al. (2015a;b), numerous other selection criteria have also been explored recently Chao et al. (2020); Sanh et al. (2020); Sehwag et al. (2020). Inspired by recent success in pruning, our proposed training procedure makes use of an importance score-based optimization approach for dropout mask optimization Ramanujan et al. (2020); Sehwag et al. (2020). There have also been efforts to compress a network at initialization, omitting pre-training Frankle et al. (2020); Hayou et al.. However, post-training pruning methods typically outperform these methods. In addition to network compression, network pruning has been used for goals like multi-task learning Mallya & Lazebnik (2018) and continual learning Golkar et al. (2019). In this work, we demonstrate pruning can

be an effective regularization strategy and can further be used to obtain a subnetwork ensemble with a performance matching that of an explicit ensemble of full networks.

**Dropout**    First introduced as a technique to regularize networks, dropout involves independent random removal of neurons or weights during training with a pre-determined probability Srivastava et al. (2014). Later, Gal and Ghahramani Gal & Ghahramani (2016) showed that dropout can be applied at test-time, called Monte Carlo (MC) dropout, which can be viewed as an approximate Bayesian technique and yields better estimates of uncertainty. Several improvements have also been proposed to improve MC dropout Antorán et al. (2020); Chen et al. (2020); Gal et al. (2017); Kendall & Gal (2017); Zhang et al. (2019). Unlike MC dropout which generates random masks on the fly at every iteration, we demonstrate that pruning-derived dropout masks can lead to significantly better performance.

## 3   Preliminaries

**Problem Setup**    We consider the problem of $k$-class classification, though the proposed method can be trivially extended to the regression setting. Suppose we are given a dataset $\mathcal{D} = \{\boldsymbol{x}_i, y_i\}_{n=1}^n$ where each feature-label pair $(\boldsymbol{x}_i, y_i) \in \mathcal{X} \times \mathcal{Y}$, and $\mathcal{X} \subseteq \mathbb{R}^d$ and $\mathcal{Y} = \{1, .., k\}$ denotes the feature space and label space respectively. Typically, a neural network (NN) $f_{\boldsymbol{w}}(\boldsymbol{x})$ parameterized by $\boldsymbol{w}$ can be used to map input features to corresponding labels for classification purpose. We define a likelihood model $p(y|\boldsymbol{x}; \boldsymbol{w}) = \mathtt{Cat}\left(\mathtt{softmax}\left(f_{\boldsymbol{w}}(\boldsymbol{x})\right)\right)$, a categorical distribution with parameters $\mathtt{softmax}\left(f_{\boldsymbol{w}}(\boldsymbol{x})\right) \in \Delta(k)$. Here $\Delta(k)$ denotes the $k$-dimensional probability simplex. Typically, maximum likelihood estimation (MLE) is performed on the train dataset to obtained the optimal parameters for the NN. At test time, $p(y|\boldsymbol{x}; \boldsymbol{w})$ is supposed to reflect the uncertainty in the predictions of the network. However, modern NNs are often poorly calibrated, yielding overly-confident predictions Guo et al. (2017).

**Deep Ensembles**    Previous work Lakshminarayanan et al. (2016) demonstrates deep ensembles, or ensembles of independently-trained NNs, as an effective remedy to the calibration problem. Given an ensemble of models (each with its parameter $\boldsymbol{w}_i$), an aggregated prediction can be obtained with

$$p(y|\boldsymbol{x}) = \frac{1}{N} \sum_{n=1}^{N} p(y|\boldsymbol{x}; \boldsymbol{w}_i). \tag{1}$$

Leveraging the under-specification property D'Amour et al. (2020) of modern neural networks together with stochasticity provided through random initialization of NNs and stochastic optimization, deep ensembles often lead to a drastic improvement in terms of both accuracy and quality of uncertainty estimates. However, a crucial shortcoming of deep ensembles is the computational overhead; an ensemble of five models would roughly cost five times more resources, including storage. This can be prohibitive in real-world applications with computational constraints.

**MC Dropout**    MC dropout can be used as an efficient alternative for ensembling. Instead of training several models independently, one can train with dropout, where a randomly sampled Bernoulli mask is applied to the weights during forward and backward propagation. At test time, an ensemble of models can then be obtained "for free" via multiple forward passes with random instantiations of the dropout masks. Compared to deep ensembles, MC dropout incurs no additional memory cost. Nevertheless, predictions produced through MC dropout are often outperformed by deep ensembles in both accuracy and uncertainty estimates.

## 4   Fixing MC Dropout

The error-ambiguity decomposition shows that the performance of an ensemble is determined by two factors: the average performance of individual models that make up the ensemble and the degree of diversity across model predictions Jiang et al. (2017); Krogh & Vedelsby (1994). Based on this perspective, the gap between MC dropout and deep ensemble can be due to two reasons. Firstly, as we see in our Ablation Study below,

individual models sampled through dropout, on average, are no better than the independently trained models in a deep ensemble. This is probably due to the common training used for all models and the reduced effective capacity of the models. Moreover, MC dropout exhibits significantly less diversity than deep ensembles, likely due to the high degree of parameter sharing between models and, again, the common training paradigm. In this section, we propose changes to MC dropout to obtain an ensemble of non-overlapping and independently trained subnetworks.

### 4.1 Toward Enhancing Model Diversity

In a standard dropout training scheme, only a very small portion of the parameters is dropped out (typically, dropout rates are set to be around 10% to 20%.). As a direct consequence, there is extensive parameter sharing among models sampled through MC dropout, leading to poor model diversity. While model diversity can be naively increased by choosing a higher dropout rate, this often results in much worse individual models, thus hampering the overall performance. Moreover, compared to models in a deep ensemble which enjoy a completely independent optimization procedure, in dropout, a random model is generated on the fly at every training iteration. This can also increase the correlation of predictions among MC dropout models.

**Orthogonal Dropout**   To enhance model diversity, instead of drawing random Bernoulli masks on the fly at each iteration, we propose to use a set of fixed, non-overlapping dropout masks during training. We term this the "orthogonal dropout." These masks can be generated by simply randomly partitioning every layer's weights into $k$ non-overlapping sets. For instance, we can randomly partition a standard neural network into an ensemble of five subnetworks each with 20% of the weights. Since the dropout masks are non-overlapping by construction, we can completely decouple the training of the subnetworks. With orthogonal dropout, each dropout subnetwork is effectively an independent model, thereby allowing us to achieve much more model diversity. To further decouple the dropout models, we can also maintain an independent set of batchnorm layers Ioffe & Szegedy (2015) for each subnetwork. Similar to MC dropout, at inference, we can apply dropout masks to weights of NNs before forward propagation and aggregate predictions of the subnetworks following Equation 1.

Unlike MC dropout which contains effectively an infinite number of models to be sampled from, with orthogonal dropout, there is a predetermined, fixed amount of dropout subnetworks for ensembling. Nevertheless, as we show in our experiments, the gain from the decoupled training procedure and resulting diversity offsets the limitation of the relatively small ensemble size.

We note that orthogonal dropout incurs some additional memory cost. Since the dropout masks are fixed before training, we need to keep track of these. Moreover, for a NN with $k$ subnetworks, we might need $k$ sets of batchnorm layers and $k$ sets of bias terms in any fully connected (FC) layers [2]. Nevertheless, compared to the number of parameters in modern architectures, this additional memory cost is negligible. The additional cost of batchnorm layers can also be mitigated through batchnorm-free NNs Brock et al. (2021). Finally, as we present below, we can share FC layers between subnetworks, which further reduces memory burden.

### 4.2 Enhancing Individual Model Performance

Naive orthogonal dropout implementation with randomly generated dropout masks can cause lackluster individual model performance (see Ablation Study below). Indeed, for an orthogonal dropout ensemble with 5 subnetworks, each dropout subnetwork contains 20% of the parameters compared to a model in a deep ensemble, likely causing an accuracy drop in individual models and hence the overall ensemble performance.

**Orthogonal Dropout Mask Optimization**   Our central goal is to maximize orthogonal dropout subnetwork performance given a predetermined level of sparsity. A natural solution to this would be to optimize the dropout masks and weights simultaneously. Let $\boldsymbol{m}_i \in \{0, 1\}^n$ for $i = 1, ..., k$ denote binary dropout

---

[2]We do not have bias terms for convolutional layers, as is common in many deep architectures

---

**Algorithm 1:** Orthogonal Dropout Optimization

---

**Input:** NN parameters $\boldsymbol{w}$; subnetwork size $k$.
**Output:** Optimized NN parameters $\boldsymbol{w}$; dropout subnetwork masks $\{\boldsymbol{m}_1, ..., \boldsymbol{m}_k\}$.
Initialize $\boldsymbol{w}_0 = \boldsymbol{w}$, $\boldsymbol{m}_0 = \mathbf{1}$ (an identity mask of all 1's) ;
**for** $i = 1 : k$ **do**
    $\boldsymbol{w}_i = \boldsymbol{w}_{i-1} \circ (\mathbf{1} - \boldsymbol{m}_{i-1})$ ;                           `// mask parameters already in use`
    Randomly initialize $\boldsymbol{w}_i$;
    Minimize $\mathbb{E}_{(\boldsymbol{x},y) \sim \mathcal{D}} \left[ \mathcal{L}\left(p(y|\boldsymbol{x}; \boldsymbol{w}_i), y\right) \right]$ w.r.t $\boldsymbol{w}_i$ ;                 `// pre-training step`
    Apply the modified `edge-popup` algorithm to find optimal $\boldsymbol{m}_i$ given $\boldsymbol{w}_i$ ;      `// pruning step`
    Minimize $\mathbb{E}_{(\boldsymbol{x},y) \sim \mathcal{D}} \left[ \mathcal{L}\left(p(y|\boldsymbol{x}; \boldsymbol{w}_i \circ \boldsymbol{m}_i), y\right) \right]$ w.r.t $\boldsymbol{w}_i \circ \boldsymbol{m}_i$ ;      `// finetuning step`
**end**

---

masks applied to $n$ weights. Then, a general optimization objective can be given by:

$$
\min_{\boldsymbol{w}, \boldsymbol{m}_1, ..., \boldsymbol{m}_k} \quad \mathbb{E}_{(\boldsymbol{x}, y) \sim \mathcal{D}} \left[ \mathcal{L}\left( \frac{1}{k} \sum_{i=1}^{k} p(y|\boldsymbol{x}; \boldsymbol{w} \circ \boldsymbol{m}_i), y \right) \right]
$$
$$
\text{s.t.} \quad \frac{n}{\|\boldsymbol{m}_i\|_0} = k \text{ and } \boldsymbol{m}_i \perp \boldsymbol{m}_k, \forall i \neq j \in \{1, ..., k\},
\tag{2}
$$

where $\circ$ denotes the Hadamard product, $\|\cdot\|_0$ denotes the number of non-zero elements in a vector, and $\perp$ indicates orthogonality (i.e., the vector product is zero). The first condition ensures that all the dropout subnetworks contain the same number of parameters while the second condition enforces the masks to be non-overlapping.

The above optimization is infeasible to solve in practice. We make two observations that allows us to propose an approximate solver. First, given the individual masks $\boldsymbol{m}_i$, the problem reduces to $k$ independent weight learning problems. Second, optimizing for a dropout mask is similar to the problem of pruning a neural network.

Based on these observations, we propose to simplify the optimization procedure into a series of greedy optimizations of $\{\boldsymbol{w}_i, \boldsymbol{m}_i\}$, for $i = \{1, \ldots, k\}$. To do this, we adopt a three-step approach originally proposed for network pruning. Specifically, at the $i$-th iteration, a pre-training step is first executed on all available model weights that exclude the ones used (i.e, not masked out) so far. Given the pre-trained parameters, an optimization step is then performed to determine the optimal dropout mask $\boldsymbol{m}_i$ before a final fine-tuning step with only the retained weights $\boldsymbol{w}_i \circ \boldsymbol{m}_i$.

The pre-training and finetuning steps are straightforward with stochastic gradient descent (SGD) given the dropout masks. On the other hand, the intermediate step of binary dropout mask optimization of $\boldsymbol{m}_i$ is worth focusing on. In this work, we adopt the score-based `edge-popup` algorithm Ramanujan et al. (2020) to find the optimal mask $\boldsymbol{m}_i$ given a pre-trained network weights. In essence, the `edge-popup` algorithm transforms the discrete optimization problem into a differentiable problem where SGD can be used. This is achieved by assigning a continuous score to each weight in the NN indicating its relative importance. During a forward pass, a binary mask $\boldsymbol{m}_i$ can be generated by ranking these scores and choosing the weight with the largest scores. A gradient for this sort and choose layer can be approximated via a relaxed backward pass. Detailed description of the algorithm can be found in the Appendix.

The original `edge-popup` algorithm is applied to a randomly initialized network Ramanujan et al. (2020). As such, the importance scores are also randomly initialized in their setup. In our work, however, the `edge-popup` algorithm is applied on a pre-trained network. In this scenario, we found it critical to initialize the scores proportional to the magnitude of weights. This is inspired by the effectiveness of the weight magnitude-based method for network pruning Frankle & Carbin (2018). A similar observation was also made previously by Sehwag et al. Sehwag et al. (2020). In practice, we use magnitudes of weights divided by the layer-wise maximum as the initial values of scores so that all scores have values in $[-1, 1]$. Another modification we applied is that, at the $i$-th iteration, weights that were used in a previous dropout iteration are masked out and excluded from consideration by the `edge-popup` algorithm. Thus, we propose to

Table 1: Results for ResNet models on various datasets. Best results for efficient ensembles are highlighted in bold. Fixed classification layer is used for orthogonal Dropout. See Table 3 and the Appendix for further ablation study on this.

| | Method | Accuracy (↑) | NLL (↓) | ECE (↓) | Size |
|---|---|---|---|---|---|
| CIFAR10 ResNet18 | Deterministic | 93.5% | 0.296 | 0.0408 | 1× |
| | MC Dropout | 94.4% | 0.191 | 0.0202 | 1× |
| | BatchEnsemble | 94.8% | 0.203 | 0.0269 | ∼1× |
| | MIMO Ensemble | 94.3% | 0.205 | 0.0180 | ∼1× |
| | Masksemble | 93.8% | 0.202 | 0.0099 | ∼1× |
| | Orthogonal Dropout (Ours) | **95.1%** | **0.157** | **0.0082** | ∼1× |
| | Deep Ensemble | 94.8% | 0.175 | 0.0110 | 5× |
| CIFAR100 ResNet18 | Deterministic | 73.0% | 1.28 | 0.141 | 1× |
| | MC Dropout | 73.3% | 1.11 | 0.0902 | 1× |
| | BatchEnsemble | 74.3% | 1.05 | 0.0910 | ∼1× |
| | MIMO Ensemble | 73.8% | 1.09 | 0.0664 | ∼1× |
| | Masksemble | 73.7% | 0.999 | 0.0224 | ∼1× |
| | Orthogonal Dropout (Ours) | **77.7%** | **0.864** | **0.0191** | ∼1× |
| | Deep Ensemble | 76.7% | 0.921 | 0.0377 | 5× |
| CUB200 ResNet50 | Deterministic | 50.1% | 2.45 | 0.196 | 1× |
| | MC Dropout | 50.2% | 2.49 | 0.171 | 1× |
| | BatchEnsemble | 54.0% | 2.33 | 0.128 | ∼1× |
| | MIMO Ensemble | 52.0% | 2.11 | 0.0879 | ∼1× |
| | Masksemble | 49.6% | 2.32 | 0.118 | ∼1× |
| | Orthogonal Dropout (Ours) | **61.4%** | **1.67** | **0.0335** | ∼1× |
| | Deep Ensemble | 55.6% | 1.98 | 0.0725 | 5× |
| Tiny-Imagenet ResNet50 | Deterministic | 56.3% | 2.18 | 0.164 | 1× |
| | MC Dropout | 56.3% | 1.96 | 0.0871 | 1× |
| | BatchEnsemble | 54.3% | 2.31 | 0.185 | ∼1× |
| | MIMO Ensemble | 54.0% | 1.96 | 0.0461 | ∼1× |
| | Masksemble | 50.1% | 2.07 | 0.0405 | ∼1× |
| | Orthogonal Dropout (Ours) | **63.2%** | **1.55** | **0.0189** | ∼1× |
| | Deep Ensemble | 61.4% | 1.73 | 0.0336 | 5× |

approximately solve the original optimization problem of Equation 2 with a greedy, sequential optimization procedure. A summary of the proposed orthogonal dropout algorithm can be found in Algorithm 1.

**Fixing The Classification Layer** Applying dropout to the final fully-connected (i.e., classification) layer can significantly reduce the performance of each subnetwork, hampering overall ensemble performance. One way around this is to not apply dropout and have all subnetworks share the classification layer. With this in mind, and inspired by some recent reports Hoffer et al. (2018); Ulyanov et al. (2018) in addition to our own empirical experience, we found randomly initializing and freezing a shared (no dropout) classification layer to be very effective. Thus, this is our default implementation for the proposed orthogonal dropout method in our experiments. Ablation study below includes results without a fixed classification layer.

## 5 Experiments

To evaluate the performance of the proposed method, we conduct extensive experiments with popular NN architectures on several benchmark datasets. We use the CIFAR-10 and CIFAR-100 dataset Krizhevsky et al.

(2009), the CUB-200 dataset Welinder et al. (2010) and the tiny-imagenet[3] dataset Deng et al. (2009). We use the ResNet18 He et al. (2016) and the Wide-ResNet28-10 Zagoruyko & Komodakis (2016) for CIFAR datasets, and ResNet50 model for the CUB-200 and the Tiny-Imagenet datasets. We use accuracy, the negative log-likelihood (NLL) Lakshminarayanan et al. (2016) and the expected calibration error (ECE) Guo et al. (2017) to measure performance. ECE, in particular, is a measure of the quality of uncertainty.

**Baselines** In addition to MC dropout and deep ensembles, we compare orthogonal dropout with several other recently proposed state-of-the-art methods for efficient ensembles. These include BatchEnsembles Wen et al. (2019), MIMO ensembles Havasi et al. (2020) and Masksembles Durasov et al. (2020). We use an ensemble of 5 models for all types of ensembling methods except MIMO, for which we found an ensemble of 2 models gave the best performance for ResNet models. Lastly, during inference, we do 30 forward passes for MC dropout which we observe was sufficient to achieve its best performance.

**Optimization** For ResNet models, we use SGD with identical hyper-parameters as originally used in the ResNet paper. We optimize the models for 150 epochs during both the pre-training and finetuning step, and optimize dropout masks for 20 epochs during pruning step for our orthogonal dropout. In order to ensure a fair comparison, we train baseline models for longer so that they are fully converged. MC dropout, MIMO ensembles and the models in deep ensembles are trained for 200 epochs respectively. When training MIMO ensembles, we also use a batch repetition of 4 to enhance the model performance, as suggested in the original paper. We empirically observe that it takes longer for BatchEnsemble and Masksembles to converge, and thus train these models for 500 epochs. For experiments using Wide-ResNet28-10, we follow the identical training procedure used by Havasi et al. Havasi et al. (2020) for a fair comparison against their experimental results.

## 5.1 Results

Experimental results for the ResNet models are summarized in Table 1. As seen clearly from the table, our proposed method significantly outperforms other recently proposed state-of-the-art (SOTA) methods of memory-efficient ensembles in terms of both accuracy and quality of uncertainty estimates. Similar trends can be also observed from experiments with Wide ResNet28-10 as well; our proposed method consistently outperforms other SOTA methods.

More interestingly, for experiments with the ResNet models, we can see from Table 1 that orthogonal dropout even produces performance better than that of standard deep ensembles, which consumes approximately five times more memory resources during inference time.This improvement is likely due to at least two reasons. Firstly, as we discuss further below, the sequential dropout mask optimization serves as an additional vehicle for regularization. Secondly, as we will further elaborate in Section 5.2, this improvement over deep ensembles is also due to fixing the classification layer. Nevertheless, we emphasize that, even without fixing the classification layer, our method consistently outperforms other recently proposed SOTA efficient ensembling techniques. This can be confirmed by comparing results for orthogonal dropout without the fixed classification layer summarized in Table 3. Lastly, we note that the relative gap between orthogonal dropout and deep ensembles becomes negligible for the experiments with Wide ResNet28-10 for CIFARs. We conjecture that this is because a higher L2 regularization is used for training of this model, following the exact training configuration of Havasi et al. Havasi et al. (2020), thereby nullifying the regularization effect of fixing the classifier layers. We leave it as future work to further understand this regularization effect.

**Individual Model Performance** To gain further insights into orthogonal dropout, we show in Figure 1 bar plots of the accuracy of the individual subnetworks in ResNet models on various datasets. Firstly, it can be seen that for all datasets except CUB200, subnetworks obtained later during the proposed greedy optimization procedure, in general, exhibit poorer performance. Intuitively, this is because later subnetworks have fewer parameters for learning. For instance, for an orthogonal dropout ensemble with five subnetworks, during training of the third subnetwork, only 60% of the weights are available for parameter and dropout mask optimization. Interestingly, we see that the 2nd model is consistently the best performing subnetwork

---

[3]https://www.kaggle.com/c/tiny-imagenet

Table 2: Results for Wide ResNet28-10. Asterisk symbol (*) represents results adapted directly from Havasi et al. (2020). Best results for efficient ensembles are highlighted in bold.

| | CIFAR10 | | | CIFAR100 | | |
|---|---|---|---|---|---|---|
| | Accuracy ($\uparrow$) | NLL ($\downarrow$) | ECE ($\downarrow$) | Accuracy ($\uparrow$) | NLL ($\downarrow$) | ECE ($\downarrow$) |
| Deterministic* | 96.0% | 0.159 | 0.023 | 79.8% | 0.875 | 0.086 |
| MC Dropout* | 95.9% | 0.160 | 0.024 | 79.6% | 0.830 | 0.050 |
| BatchEnsemble* | 96.2% | 0.143 | 0.021 | 81.5% | 0.740 | 0.056 |
| MIMO* | 96.4% | 0.123 | 0.010 | 82.0% | 0.690 | 0.022 |
| Masksembles | 94.6% | 0.173 | 0.008 | 76.7% | 0.843 | **0.015** |
| Orthogonal Dropout (Ours) | **96.6%** | **0.122** | **0.005** | **82.8%** | **0.701** | 0.021 |
| Deep Ensembles* | 96.6% | 0.114 | 0.010 | 82.7% | 0.666 | 0.021 |

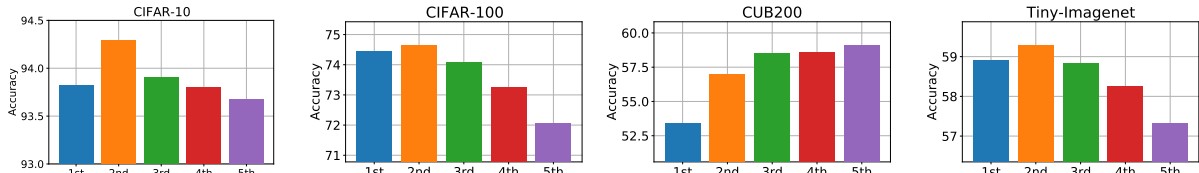

Figure 1: Bar plots of accuracy of individual orthogonal dropout subnetworks of ResNet models. "i-th" model represents the i-th subnetwork obtained using Algorithm 1 sequentially.

out of the ensemble of five subnetworks. We hypothesize that removing a small portion of parameters in neural networks can implicitly regularize a network and account for these results. This could also help explain why for the CUB200 dataset, even the 5th model outperformed the 1st by far. Compared to the other three datasets, the CUB200 dataset is significantly smaller in size, consisting of only approximately 6000 images for training. As such, aggressive regularization can potentially significantly improve generalization performance. Lastly, we note that, despite the significantly lower accuracy of later subnetworks, we found including them in the ensemble still leads to a positive gain.

## 5.2 Ablation Study

In this section, we conduct additional experiments using CIFAR10/100 and the ResNet18 model to decompose the contribution of each component of our proposed method. Specifically, we compare standard MC dropout against 1. dropout with randomly generated and orthogonal masks, 2. orthogonal dropout with mask optimization, 3. orthogonal dropout with both mask optimization and fixed classification layer. To further demonstrate the effect of fixing the classification layer, we also train a deep ensemble with a fixed classification layer. In addition to accuracy, NLL and ECE, we also report the individual level accuracy and compute the Inter-rater Agreement (IA) Kuncheva & Whitaker (2003) between individual models in an ensemble, as a measure of diversity in the ensemble to gain further insights. The lower the IA, the more diverse the ensembles.

Results for this ablation study can be found in Table 3. Firstly, we note that even orthogonal dropout without dropout mask optimization outperforms standard MC dropout significantly for CIFAR100, despite the significantly smaller ensemble size (for MC dropout, results are obtained with 30 forward passes whereas orthogonal dropout ensemble contains only five subnetworks). This can be explained by increased diversity among models in ensembles, as evident from considerably lower IA for orthogonal dropout. In fact, the amount of diversity in orthogonal dropout is almost identical to that of deep ensembles.

Furthermore, we see that optimizing dropout masks and fixing the classier layer further boost the performance of orthogonal dropout substantially. The improvement primarily is attributable to the increase in individual model performance. To isolate the effect of fixing the classifier layer, we also train a deep ensemble of 5

Table 3: Ablation study of the proposed method. orthogonal dropout methods are trained without dropout mask optimization. "MO" corresponds to "mask optimization" and "FC" corresponds to "Fixed Classifier". "Ind Acc" denotes the averaged individual model accuracy in an ensemble, while "Ens Acc" represents the ensemble accuracy.

| | CIFAR10 | | | | | CIFAR100 | | | | |
|---|---|---|---|---|---|---|---|---|---|---|
| | Ind Acc ($\uparrow$) | Ens Acc ($\uparrow$) | NLL ($\downarrow$) | ECE ($\downarrow$) | IA ($\downarrow$) | Ind Acc ($\uparrow$) | Ens Acc ($\uparrow$) | NLL ($\downarrow$) | ECE ($\downarrow$) | IA ($\downarrow$) |
| MC Dropout | 93.4% | 94.4% | 0.287 | 0.0415 | 0.708 | 71.3% | 73.3% | 1.11 | 0.0902 | 0.776 |
| Orthogonal Dropout | 93.2% | 94.3% | 0.188 | 0.0138 | 0.597 | 71.4% | 75.1% | 0.977 | 0.0455 | 0.655 |
| Orthogonal Dropout+MO | 93.8% | 94.9% | 0.176 | 0.0122 | 0.601 | 72.3% | 76.3% | 0.935 | 0.0328 | 0.648 |
| Orthogonal Dropout+MO+FC | 93.9% | 95.1% | 0.157 | 0.0082 | 0.594 | 73.7% | 77.7% | 0.864 | 0.0191 | 0.638 |
| Deep Ensemble | 93.4% | 94.8% | 0.175 | 0.0110 | 0.581 | 72.7% | 76.7% | 0.921 | 0.0377 | 0.652 |
| Deep Ensemble+FC | 93.5% | 95.1% | 0.151 | 0.0091 | 0.580 | 74.1% | 77.8% | 0.858 | 0.0221 | 0.645 |

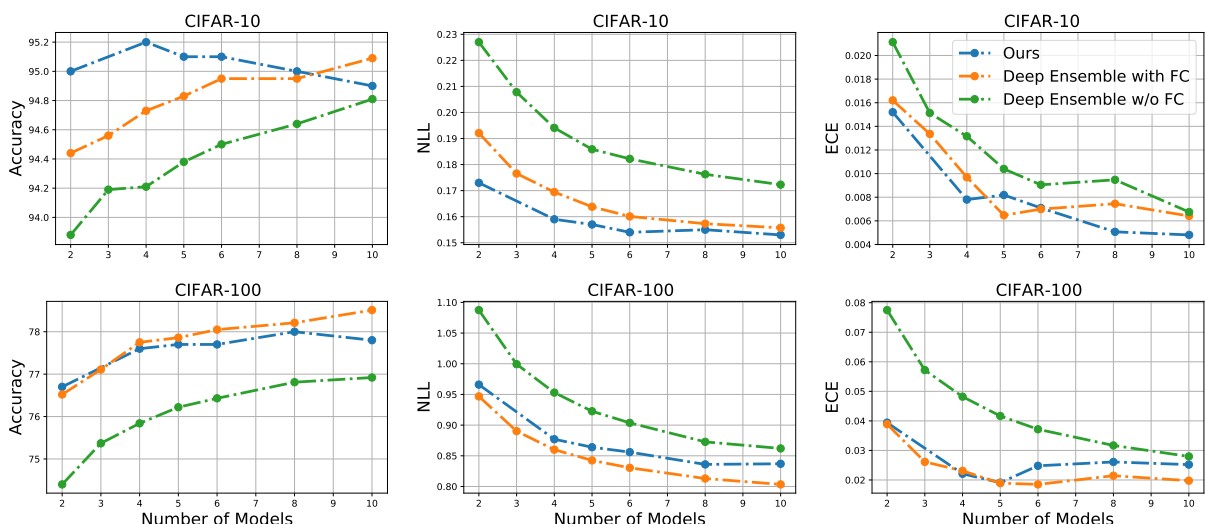

Figure 2: Plot of accuracy/NLL/ECE against number of models in the ensembles. For orthogonal dropout, number of models is varied by changing the size of each subnetwork and all the orthogonal dropout ensembles are of the same size. "FC" corresponds to "Fixed Classifier".

models with a fixed classifier layer. Note that fixing the classification layer consistently yields an ensemble with better performance in this particular experimental setup. Remarkably, orthogonal dropout models with mask optimization and fixed classifier layer perform as well as deep ensembles with 5 models with fixed classifier layer in terms of both accuracy and calibration.

## 5.3 How Many Subnetworks Can We Fit?

We also investigate how many subnetworks we can fit into a ResNet18 model. This can be achieved by adjusting the percentage of parameters each subnetwork consumes. For instance, if each subnetwork consists of 50% of the parameters, we can fit in 2 subnetworks into a ResNet18 model, and if each subnetwork consists of 10% of the parameters, the orthogonal dropout ensemble would contain 10 subnetworks. Nevertheless, increasing the ensemble size can decrease the quality of individual model performance. As such, there is an inherent trade-off that needs to be balanced.

In figure 2, we plot model performance (ensemble accuracy, NLL and ECE) against the number of subnetworks in a single ResNet18 model. Surprisingly, we see that we can even fit 10 subnetworks into a ResNet18 model using orthogonal dropout with an extremely competitive ensemble performance. This is in stark contrast with the MIMO ensemble, which also aims at fitting multiple subnetworks into one model, whose performance degrades drastically with 4 or more subnetworks in the model Havasi et al. (2020). Moreover,

Table 4: Comparison against deep ensembles with reduced convolutional kernel size, so that deep ensemble has the same number of parameters as orthogonal dropout. "FC" stands for fixed classification.

| | CIFAR10 | | | CIFAR100 | | |
|---|---|---|---|---|---|---|
| | Accuracy (↑) | NLL (↓) | ECE (↓) | Accuracy (↑) | NLL (↓) | ECE (↓) |
| rescaled Ensemble + FC | 94.6% | 0.174 | 0.0104 | 76.0% | 0.911 | 0.0256 |
| Orthogonal Dropout + FC | 95.1% | 0.157 | 0.0082 | 77.7% | 0.864 | 0.0191 |

Table 5: Comparison against baseline methods when all methods have fixed classification layer.

| | CIFAR10 | | | CIFAR100 | | |
|---|---|---|---|---|---|---|
| | Accuracy (↑) | NLL (↓) | ECE (↓) | Accuracy (↑) | NLL (↓) | ECE (↓) |
| Dropout + FC | 94.5% | 0.187 | 0.0185 | 73.6% | 1.12 | 0.0863 |
| Batch Ensemble + FC | 94.6% | 0.210 | 0.0303 | 74.6% | 1.02 | 0.0898 |
| MIMO + FC | 94.6% | 0.182 | 0.0146 | 75.1% | 0.988 | 0.0384 |
| Masksemble + FC | 93.5% | 0.203 | 0.0090 | 73.6% | 0.969 | 0.0143 |
| Orthogonal Dropout + FC | 95.1% | 0.157 | 0.0082 | 77.7% | 0.864 | 0.0191 |

WRN28-10, a network with much more parameters, was used in their experiment. With ResNet18, we empirically observe even 3 subnetworks in MIMO yields a poor performance (see Appendix).

To further understand the quality of orthogonal dropout ensemble, we also plot the performance achievable by an explicit deep ensemble with the same numbers of models in the ensemble. In general, we see that orthogonal dropout is capable of outperforming even a standard deep ensemble of 10 models for this particular experimental setup. Moreover, we see that orthogonal dropout matches a deep ensemble of 8 models with fixed classification layers, giving us significant memory saving.

### 5.4 Additional Comparisons against Deep Ensembles

To further demonstrate the effectiveness of the proposed method, we conduct an additional comparison against an ensemble of 5 independently trained networks using ResNet, but each with convolutional filters reduced by a factor of $\sqrt{1/5}$, so that in total, the size of this explicit ensemble is the same as that of an orthogonal dropout model. We report the comparison in Table 4. As seen clearly, our proposed method is capable of significantly outperforming it.

### 5.5 Additional Experiments with Fixed Classifier Layer

To further demonstrate that fixing the classification layer is not the main source of increase in performance, we conduct an additional experiment and fix the classification layer for all baseline methods using ResNet-18 and CIFAR datasets. Results of the experiments are summarized in Table 5. The proposed method significantly outperforms all other SOTA methods even when the classifier layers are fixed for all methods.

### 5.6 Additional Comparisons against Dropout

We also conduct additional ablation study to compare against other forms of more advanced dropout methods like DropChannel and DropBlock. Results of the experiments are summarized in Table 6. To summarize, orthogonal dropout outperforms all alternative forms of dropout in both CIFAR10 and CIFAR100 tested consistently.

## 6 Discussion

In this work, we proposed orthogonal dropout, an easy-to-implement technique that allows us to split a single high capacity neural network model into an ensemble of subnetworks. In our experiments, we demon-

Table 6: Comparison against other dropout baselines.

| | CIFAR10 | | | CIFAR100 | | |
|---|---|---|---|---|---|---|
| | Accuracy (↑) | NLL (↓) | ECE (↓) | Accuracy (↑) | NLL (↓) | ECE (↓) |
| Dropout | 94.4% | 0.191 | 0.0202 | 73.3% | 1.11 | 0.0902 |
| DropChannel | 94.3% | 0.239 | 0.0343 | 73.9% | 1.21 | 0.136 |
| DropoutBlock | 94.5% | 0.163 | 0.0139 | 73.3% | 1.226 | 0.137 |
| Orthogonal Dropout | 95.1% | 0.157 | 0.0082 | 77.7% | 0.864 | 0.0191 |

strate and discuss the regularization effect achieved by training pruned networks and using a randomized and frozen fully connected layer in the network. Finally, we present exhaustive results that show that our method consistently outperforms several of the recently proposed state-of-the-art methods for efficient ensembles. Furthermore, our method achieves accuracy and uncertainty values matching that of an explicit deep ensemble, while demanding significantly less storage.

Lastly, we describe several shortcomings of the proposed method for potential future exploration. Firstly, as we have demonstrated in Figure 1, there is significant variation in performance across the subnetworks obtained. This could potentially be harmful to the performance of the ensemble. In addition, since dropout subnetworks are trained one by one, there is no training time reduction compared to training of deep ensembles. As such, we hope to explore alternative frameworks for dropout subnetworks optimization. Moreover, unlike the MIMO ensemble, our proposed method requires multiple forward passes. This problem can be potentially solved by instead saving the dropout subnetworks independently.

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

## A   A Brief Review of the `edge-popup` Algorithm

We give a brief review of the `edge-popup` algorithm in this section. For simplicity, we describe the algorithm with a fully connected neural network. The algorithm can be easily extended to the case of CNNs.

Suppose we have an $L$-layer fully connected NN with parameters $\boldsymbol{w} = \{W^{(1)}, ..., W^{(L)}\}$. If we let $\boldsymbol{x} = \boldsymbol{x}^{(0)}$ to be the input to the NN and $\boldsymbol{x}^{(h)}$ to be the $h$-th hidden layer of the NN, then a standard NN can be defined recursively by

$$\boldsymbol{x}^{(h)} = \sigma\left(W^{(h)}\boldsymbol{x}^{(h-1)}\right), \quad 1 \leq h \leq L,$$

where $\sigma$ denotes some non-lienar activations functions like the ReLU activation function.

Now, in order to select a subset of weights from $\boldsymbol{w}$, for each weight in the parameters $\boldsymbol{w}$, we learn a popup score associated with it. We denote the popup scores by $\boldsymbol{s} = \{S^{(1)}, ..., S^{(L)}\}$. Note that, each score matrix $S^{(h)}$ is of the same dimension as that of $W^{(h)}$. Then, given the set of score matrices, a set of binary masks $\boldsymbol{m} = \{M^{(1)}, ..., M^{(L)}\}$ can be generated. Specifically, for each score matrix $S^{(h)}$, we sort the popup scores based on magnitude of the scores at each layer. With a pre-determined ratio $k\%$, $M_{ij}^{(h)} = f\left(S_{ij}^{(h)}\right) = 1$ if $\left|S_{ij}^{(h)}\right|$ is among the top $k\%$ highest scores in the $h$-th layer, and $M_{ij}^{(h)} = 0$ otherwise. Then, during the forward pass of NN with the `edge-popup` Algorithm, binary masks are applied onto the weight matrices before the forward propagation

$$\boldsymbol{x}^{(h)} = \sigma\left(\left(M^{(h)} \circ W^{(h)}\right)\boldsymbol{x}^{(h-1)}\right), \quad 1 \leq h \leq L,$$

where $\circ$ denotes the Hadamard product.

During the entire learning procedure of the `edge-popup` Algorithm, the weight matrices stay fixed, and only the score matrices are updated with gradient descent. Note that, due to the use of binary masks, direct computation of the gradient is impossible. As such, the straight-through gradient estimator is used instead so that the thresholding function $f(\cdot)$ is replaced by the identity function instead. This allows us to approximate the gradient for $S_{ij}^{(h)}$ by

$$\frac{\partial \mathcal{L}}{\partial \boldsymbol{x}^{(h)}} \frac{\partial \boldsymbol{x}^{(h)}}{\partial S_{ij}^{(h)}} = \frac{\partial \mathcal{L}}{\partial \boldsymbol{x}^{(h)}} W_{ij}^{(h)} \boldsymbol{x}_i^{(h-1)},$$

where $\mathcal{L}$ denotes the cross-entropy loss. Given the gradient estimator, the popup scores can then be updated via stochastic gradient descent.

Lastly, we note that a naive random initialization of popup scores as proposed by Ramanujan et al. Ramanujan et al. (2020) can lead to significantly worse performance. To this end, we instead choose to initialize the popup scores based on the weights of the trained NNs. this is inspired by the recent success of magnitude-based pruning techniques. As such, for each layer $h$, we initialize the scores by

$$S_{ij}^{(h)} = \frac{W_{ij}^{(h)}}{\max(|W^{(h)}|)},$$

where $\max(|W^{(h)}|)$ denotes the maximum magnitude of the matrix $W^{(h)}$ so that all popup scores are normalized between $[-1, 1]$. Similar approach was also adopted by Sehwag et al. Sehwag et al. (2020).

## B   Additional Ablation Studies

**Optimal Number of Subnetworks in MIMO Networks**   We experimentally investigate the number of subnetworks that can be fit into a MIMO network with the ResNet18 model using CIFAR10 and CIFAR100. We use the identical training procedure for all models as described in Section 5, varying only the number of subnetworks, as a direct comparison against our proposed orthogonal dropout method. Note that for MIMO

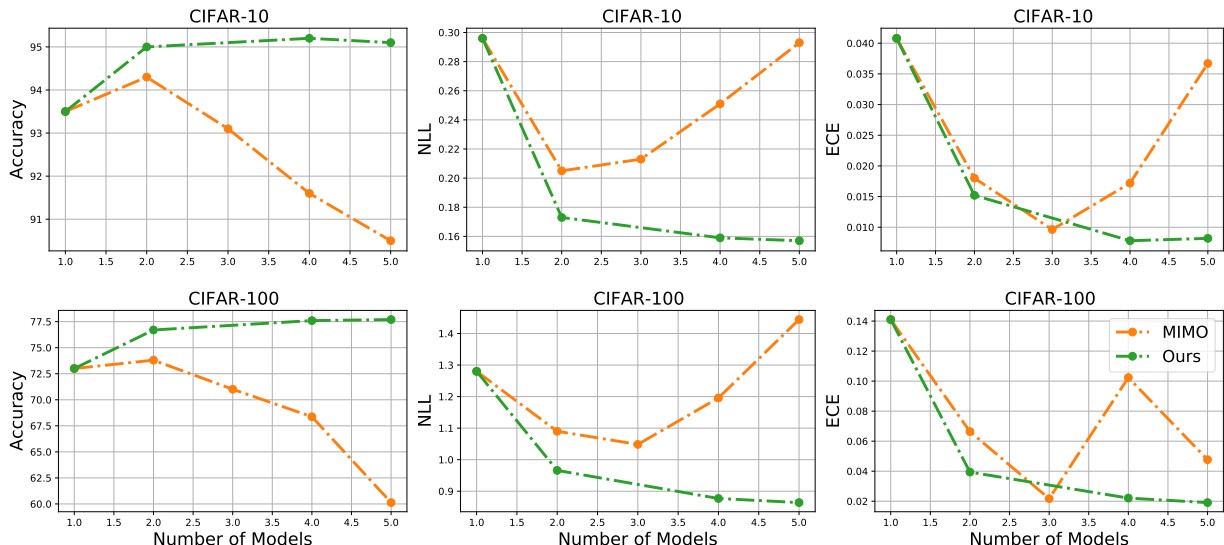

Figure 3: Plot of accuracy/NLL/ECE against number of models in the ensembles. For the proposed method, the number of models is varied by changing the size of each subnetwork and all the orthogonal dropout ensembles are of the same size. For MIMO networks, the number of models is varied by changing the number of inputs and outputs (classifier layers) of the networks.

models, changing the number of subnetworks amounts to simply adjusting the number of input images and the number of linear classification layers; three subnetworks correspond to a network with three inputs and three outputs.

Results are summarized in Figure 3. A direct comparison of MIMO networks against our proposed orthogonal dropout strategy reveals that we can fit much more models into a network of the same capacity. Indeed, for the ResNet18 model, having even three subnetworks in the MIMO networks can significantly degrade accuracy performance. We hypothesize that this is due to the way subnetworks are implemented in MIMO networks. During the training of MIMO networks, multiple inputs are concatenated together and fed into the networks simultaneously. When the size of the subnetworks grows, the number of channels in the concatenated inputs also grows proportionally, thereby making the simultaneous training of the subnetworks harder. After all, each input in the stack of inputs is independent of one another. Yet, there is no explicit constraint/regularization in MIMO networks to enforce such independence. As such, when the number of subnetworks becomes large, it can be hard for networks to capture such independence between the input images by themselves.

