# OpenReview forum: "Ex Uno Plures: Splitting One Model into an Ensemble of Subnetworks"
_TMLR — Rejected by TMLR_

### Review · Reviewer_xWwx · 2022-07-21

**Summary Of Contributions:**

Proposes efficient ensembling approach for NNs. Relative to most closely related work, Durasov et al. 2020, proposes optimization strategy for minimizing parameter sharing between ensemble members, by borrowing recent contributions from the pruning literature, namely Ramanujan et al. (2020).

**Broader Impact Concerns:**

No concerns.

**Requested Changes:**

No notable issues observed, clearly introduced contribution with clear, interesting formulation that appears to perform well relative to strong  baselines. With that said, not an expert on the efficient ensembling - uncertainty estimation literature.

**Strengths And Weaknesses:**

Strengths:
- Well-motivated & concepts introduced sequentially in a digestible manner.

Weaknesses:
- Requires further proofreading, e.g. "Lakshminarayanan et al. Lakshminarayanan et al. (2016)... Sinha et al. Sinha et al. (2020)... Jain et al. Jain et al. (2020)...Gal and Ghahramani Gal & Ghahramani (2016)...Sehwag et al. Sehwag et al. (2020)...Havasi et al. Havasi et al. (2020)...fixing the classier layer"
- "The above optimization is infeasible to solve in practice.", presenting the argument that led the authors to this conclusion would be helpful to the reader.
- " To do this, we adopt a three-step approach originally proposed for network pruning." The authors should provide the reference in which the original proposal was presented.
- "We also conduct additional ablation study to compare against other forms of more advanced dropout methods
like DropChannel and DropBlock." Citations missing.

---

> ### Author Response · Authors · 2022-08-08
> **Response to Comments**
>
> We want to thank the reviewer for taking the time to read our paper and for offering constructive feedback. We address the concerns raised by the reviewer below.
>
> “Requires further proofreading”
>
> - We provide a brief overview of the prior related literature to facilitate the proof-reading process in the related works section of the paper. As a brief summary, Lakshminarayanan et al. (2016) demonstrated the effectiveness of ensembles of neural networks that are trained independently. Despite its effectiveness, an ensemble of neural networks can be computationally expensive. In the meantime, another work by Gal & Ghahramani (2016) demonstrated that dropout can be used to obtain an ensemble of predictions without incurring any memory cost. This is done by sampling through dropout at test time. Nevertheless, ensembles obtained can lack diversity, and do not perform as well as explicit ensembles of independently trained neural networks. In this work, we hope to fill this gap. Specifically, we leverage recently proposed methods of network pruning to obtain an ensemble of networks while incurring minimal additional memory.
>
> "The above optimization is infeasible to solve in practice."
>
> - The optimization is infeasible to solve in practice because we are dealing with very deep and over-parameterized neural network models and constraint discrete optimization of the binary masks on top of these neural network models. To the best of our knowledge, there is not such an efficient algorithm for us to solve such an optimization problem. Nevertheless, what we propose in the paper can be seen as a greedy iterative approach to try to find an approximate solution to the optimization objective.
>
> Lastly, We will also incorporate all other minor suggestions and reflect the changes in the updated version of the manuscript.

---

### Review · Reviewer_grbB · 2022-07-25

**Summary Of Contributions:**

This paper proposes Orthogonal Dropout as a way to combine Monte Carlo (MC) Dropout with network pruning. This method provides a tradeoff between MC Dropout (which ensembles the same trained network with multiple random dropout masks) and Deep Ensembles (which ensembles the same architecture multiple times learned with different random training initialization). Instead this approach learns a sequence of subnetworks with very few shared parameters during training, and then uses the ensemble of subnetworks during inference. Empirical results show that Orthogonal Dropout has better efficiency than Deep Ensembles and higher classification accuracy than other efficient ensembling methods.

**Broader Impact Concerns:**

This paper does not raise immediate concerns about broader impact. In addition to ECE, the authors could discuss how Orthogonal Dropout compares to other ensemble methods wrt fairness and/or interpretability.

**Requested Changes:**

### Critical
- Discuss and/or run experiments to compare quantitative cost of various methods during inference _and training_ in terms of compute, memory, running time, and parallelization.
- Resolve/explain the inconsistency between rows 6,7,13,14 of Table 1 + rows 4,6 of Table 3, and row 2 of Table 4 + rows 5,2 of Table 5. These all appear to be ResNet-18 ensembles on CIFAR10/100 datasets.


### Non-critical
- Ablation in Section 5.2: It would be better to split mask optimization into pruning and finetuning to see how each step in Algorithm 1 affects performance. Then Table 3 could be Pre-train only, Pre-train + Prune, and Pre-train + Prune + Finetune.
- Typos
	- Table 2: On CIFAR100, MIMO should be bold because it has lower NLL of 0.690 compared to Orthogonal Dropout NLL of 0.701.
	- Section 5.2: "classier layer" should be "classifier layer".
	- Appendix B: "much more models" should be "many more models".


**Strengths And Weaknesses:**

### Strengths
- Simple idea using a natural combination of previous work (edge-popup pruning, deep ensembles).
- Thorough experiments show a novel tradeoff wrt previous state of the art.
- Detailed ablation study, including adding fixed FC layers to the other baselines and fixing the total number of parameters in the ensemble.

### Weaknesses
- "Cost" of various methods is only described approximately/qualitatively rather than quantitatively. The paper would be stronger if the Table 1, Table 2, Figure 2, etc. included quantitative comparisons of efficiency such as model size in disk or wall-clock inference time. Without emphasizing this, it appears as though adding FC to the Deep Ensembles baseline allows it to match the performance of Orthogonal Dropout.
- The authors compare the cost of Orthogonal Dropout to other methods using network size because this is the main constraint during model inference. The authors should also comment on the cost during training, and run additional experiments where applicable.
- Small inconsistencies in experimental results (see below).

---

> ### Author Response · Authors · 2022-08-08
> **Author Response to Reviewer grbB**
>
> We want to thank the reviewer for taking the time to read our paper and for offering constructive feedback. We are encouraged by the strengths that the reviewer has identified, and address the concerns raised by the reviewer below.
>
> “Computational Cost” - We address the concerns regarding computational costs below, and will update the manuscript accordingly to address the concerns.
>
> - Training Cost: While the proposed method inevitably takes longer to train than a single deterministic model and the MC dropout (since we are essentially training 5 models), we emphasize that, in order to ensure a fair comparison, we train MC dropout for longer so that the model has fully converged (with 99.9% training accuracy). Moreover, the other benchmark methods for efficient ensemble techniques like "MIMO", "batchEnsemble" and "Masksembles", which according to the original papers and based on our experience, all take significantly longer to train than MC dropout or a single deterministic model. We empirically observe that the time it takes to train "MIMO", "batchEnsemble" and "Masksembles" are all about the same as the proposed method with 5 subnetworks. Lastly, when compared to the deep ensembles, we train each individual model for a less amount of epochs so that the overall number of epochs Pre-train + Prune + Finetune are the same as training one model in the deep ensembles, in order to have a fair comparison. Some of these training details were omitted for the baseline methods in the submitted manuscript. We will include a detailed time taken for training for each method in the Appendix in the updated manuscript.
> - Memory Cost: We illustrate the memory cost of the proposed method using a ResNet18 model. The overall size of the saved parameter for one standard model is 56MB. Compared to that, the size of its orthogonal dropout counterpart with 5 subnetworks is 99MB. As such, each set of batchnorm and binary masks in this case is about 1/6th of the size of its original model. Compared to an explicit ensemble of 5 models, which consumes roughly 300MB, the proposed method incurs significantly less memory. There are also several venues for improvements. For instance, instead of using separate batchnorm layers, we can use one shared layer for all subnetworks (train the batchnorm layer for the first subnetwork, then freeze it for all the rest of all subnetworks). We experimentally observed that this leads to slightly worse performance in general, but is still capable of outperforming many of the efficient ensemble baselines. We will compile and include the memory costs for all of the models in the Appendix of the paper in the updated manuscript.
> - Runtime Cost: While the proposed method is capable of saving memory when compared to that of the deep ensemble, we have to admit that the proposed method does not give us an edge over regular deep ensembles in terms of runtime cost at test time, because getting an ensemble prediction would still require us to forward pass 5 times.
> - Parallelization: while parallel training is impossible at the model level, the proposed method can still benefit from parallel training at the mini-batch level, which will significantly speed up the training of each subnetwork. At inference (test) time, parallelism can be enabled by explicitly saving sparse/pruned models instead so that the memory advantage can be preserved. With an ensemble of pruned models (with a significantly less number of parameters), the runtime of each pruned model can be faster than that of a regular-sized model, depending on data structure and implementation.
>
> “Inconsistent Experiments”
>
> - We apologize for any confusion in the results section, but the inconsistency is caused by the way “orthogonal dropout” is defined in each table. In Tables 1 and 2, “orthogonal dropout” corresponds to the proposed method including “mask optimization” and “fixed classifier”, and “deep ensemble” corresponds to the method originally proposed by Lakshminarayanan et al. (2016). In Tables 3, 4, and 5, to further understand the effect of “fixed classifier”, we also incorporate the “fixed classifier” method into the baseline methods. As such, row 4 of Table 3 would correspond to rows 6 + 13 in Table 1, and row 5 of Table 3 would correspond to row 7 + 14 in Table 1.
>
> “Additional suggestion on Ablation Studies”
>
> - We will incorporate the reviewer’s suggestion and include an ablation study to show an additional result of Pre-train only, Pre-train + Prune, and Pre-train + Prune + Finetune.
>
> Lastly, We will also incorporate all other minor suggestions and reflect the changes in the updated version of the manuscript.

---

> > ### Comment · Reviewer_grbB · 2022-08-15
> > **Update Manuscript**
> >
> > Thank you for the detailed response. Is it possible to upload a revised manuscript which incorporates all of these proposed changes?

---

### Review · Reviewer_sS84 · 2022-07-27

**Summary Of Contributions:**

This paper proposes a method to compute an ensemble of subnetworks using dropout masks computed via a pruning strategy and trained independently. In the experimental analyses, the proposed method is compared with baseline ensembles and dropout methods on several computer vision datasets like CIFAR10/100, CUB200, and Tiny-Imagenet. In most of the results, the proposed method performs on par with the baseline methods, and outperforms them in several others.

The proposed method is interesting. However, there are several major and minor problems with the paper, method and analyses.

**Broader Impact Concerns:**

N/A.

**Requested Changes:**


The major problems with the paper are as follows:

1. Weighted aggregation of multiple models have been studied in the literature. Even in the recent frameworks such as federated learning, several dropout based methods have been proposed. For a reference, you can check the following paper for model aggregation and most recent related works, like ordered and nested dropouts:
https://openreview.net/forum?id=4fLr7H5D_eT

Therefore, the proposed method should be compared with additional methods.

2. In most of the analyses, the proposed method performs on par with the baseline. Indeed, the accuracy of proposed method and baseline ensemble methods are very close to each other on Cifar datasets. To highlight the difference and show the superiority of the proposed method more clearly, additional descriptive statistics and even confidence intervals should be presented. In addition, the methods should be examined on additional larger scale datasets such as Imagenet and additional network architectures.

3. One of the main claims of the paper is improved efficiency. However, first the "efficiency" was not defined clearly and precisely. Second, this claim should be analyzed in detail. For instance, memory consumption, inference and training times of the methods should be examined in comparative analyses.

4. The proposed algorithm sequentially aggregates models and optimize parameters and masks at each aggregation step. How does the result and convergence rates change as the order of model aggregation changes? Please also provide more detailed convergence analyses.

**Strengths And Weaknesses:**

The paper is well written in general. However, there a few typos and redundant notation. For instance, k denotes both number of classes and models.

The main weaknesses of the paper are insufficient comparison with related work, incomplete analyses of the experimental results and ablation studies, and insufficient justification of some of the claims. More detailed comments are provided in the next section.

---

> ### Author Response · Authors · 2022-08-09
> **Response to Reviewer sS84**
>
> We want to thank the reviewer for taking the time to read our paper and for offering constructive feedback. We address the concerns raised by the reviewer below.
>
> “Weighted aggregation of multiple models has been studied in the literature”
>
> - We thank the reviewer for pointing out this interesting line of research on federated learning. This is indeed a similar line of research that we were not aware of before. We will incorporate this line of research in the related works section of the updated manuscript. Nevertheless, we do not believe it is necessary to further benchmark our method against these methods like the Ordered Dropout proposed by [1]. Specifically, FjORD proposed by [1] is “a framework for federated training over heterogeneous clients” that “enables the extraction of lower footprint sub-models without the need of retraining.” As such, the focus is on being able to obtain individual models of various sizes (which is enabled by sampling through the different amounts of dropout) so as to be applied to devices of various computational capabilities. In sharp contrast, in our application, the focus is on obtaining an ensemble of models at no extra cost. Because of this distinction in the application domain, naively applying FjORD for NN ensembling will lead to suboptimal performance compared to ours. Indeed, because of the way Ordered Dropout is implemented, there will be a drastic variation in the performance of each individual model (See Figure 3 of [1] for example, varying from 91% to 94.5% for CIFAR10). While the proposed method of orthogonal dropout also inevitably leads to variation in the performance of individual models, such variation is much smaller (See Figure 1 of the manuscript, varying from 93.5% to 94.2% for CIFAR10). Moreover, based on the results shown in Figure 3 of [1], the performance of subnetworks is always lower than a single independently trained model of the same size. A natural conclusion from this observation is that an ensemble of FjORD sub-networks will perform worse when compared to deep ensembles. In contrast, we show in this paper that the proposed method of orthogonal dropout is capable of performing as well as deep ensembles. We believe this enhanced performance is enabled by a sequential training process, a network pruning procedure, and a regularization effect from fixed classifier layers.
>
> “additional descriptive statistics and even confidence intervals should be presented”
>
> - We agree with the reviewer and see the importance of including error bars in results. However, we also would like to emphasize that the consistent pattern of results we have obtained across datasets increases our confidence in the presented conclusions. Moreover, training ensembles of five models can be quite computationally expensive, and repeating the experiments numerous times can take up a lot of resources. That said, we have launched two more repeat experiments that will allow us to include error bars in the final publication. Preliminary results of the repeated experiments suggest that the pattern observed is consistent. Despite some minor fluctuations in numbers, our proposed method still outperforms all benchmark efficient ensemble methods.
>
> “methods should be examined on additional larger scale datasets such as Imagenet and additional network architectures.”
>
> - We draw our attention to the fact that we have tested the proposed method with the ResNet architecture of varying depths and also the wide-ResNet architecture. Consistent patterns were observed for most of our experiments conducted. We agree with the reviewer and acknowledge that the experiments can be expanded to larger datasets such as ImageNet and other architectures like DenseNet, but we believe we have conducted a thorough analysis of a diverse group of datasets and our results give us confidence in the merit of the proposed method. We would like to note that the presented experiments consumed a substantial amount of compute resources and, due to limited computational resources and time constraints, we leave any further large-scale experimentation to future work where we plan to focus on some intriguing open questions and hypotheses/ideas prompted by this work.
>
> “Computational Cost/Efficiency”
>
> - Please see the response to Reviewer grbB for a detailed response on the efficiency of the proposed method.
>
> Lastly, We will also incorporate all other minor suggestions and reflect the changes in the updated version of the manuscript.
>
> [1] Horvath, Samuel, et al. "Fjord: Fair and accurate federated learning under heterogeneous targets with ordered dropout." Advances in Neural Information Processing Systems 34 (2021): 12876-12889.

---

### Decision · Action_Editors · 2022-09-06

**Recommendation:** Reject

**Comment:**

Two reviewers felt that the author response was not satisfactory, and that the most important acceptance criterion for TMLR was not met, namely "claims made in the submission supported by accurate, convincing and clear evidence". Specifically, one reviewer commented

> "In the rebuttal, authors addressed some of the questions raised by reviewers. However, their response was mainly highly level discussion of the reviewer comments, and they **did not provide the detailed and large scale analyses required to explore major claims** and critical issues of the work and the paper."

While another reviewer commented

> "In their responses, the authors addressed concerns to some extent: citing connections to federated learning, adding error bars, quantifying cost/efficiency, additional ablations. However, **some responses lack detail** (e.g. "Preliminary results of the repeated experiments suggest that the pattern observed is consistent."). They also did not update the manuscript to reflect these changes."

I would recommend the authors fix these issues if they wish to resubmit.